# Optimal Sensor Placement in Reduced-Order Models Using Modal Constraint Conditions

**DOI:** 10.3390/s22020589

**Published:** 2022-01-13

**Authors:** Eun-Taik Lee, Hee-Chang Eun

**Affiliations:** 1Department of Architectural Engineering, Chung-Ang University, Seoul 06974, Korea; etlee@cau.ac.kr; 2Department of Architectural Engineering, Kangwon National University, Chuncheon 24341, Korea

**Keywords:** optimal sensor placement, constraint, Fisher information matrix, efficient independence, slave mode, master mode, target mode

## Abstract

Sensor measurements of civil structures provide basic information on their performance. However, it is impossible to install sensors at every location owing to the limited number of sensors available. Therefore, in this study, we propose an optimal sensor placement (OSP) algorithm while reducing the system order by using the constraint condition between the master and slave modes from the target modes. The existing OSP methods are modified in this study, and an OSP approach using a constrained dynamic equation is presented. The validity and comparison of the proposed methods are illustrated by utilizing a numerical example that predicts the OSPs of the truss structure. It is observed that the proposed methods lead to different sensor layouts depending on the algorithm criteria. Thus, it can be concluded that the OSP algorithm meets the measurement requirements for various methods, such as structural damage detection, system identification, and vibration control.

## 1. Introduction

In the finite element analysis model, it is impractical to measure the response at a full set of degrees of freedom (DOFs) for electromagnetic induction. Measurements must be undertaken using sensors that are smaller than the system order. Optimal sensor design involves establishing an appropriate number of sensors and sensor positions to investigate the structural performance. Measurements and data analysis are performed for structural health monitoring and damage detection and control, among others. Therefore, sensor locations must be designed in accordance with this purpose. Studies on optimal sensor placement (OSP) have been conducted over the past several decades. With the advent of novel measurement technologies and innovative sensors, OSP techniques have been gradually developed for more accurate evaluation.

The OSP algorithms investigate the relative influence of the signal at the sensor locations of a finite element model with a limited number of sensors. Chang and Pakzad [1] discussed and compared several OSP techniques.

The effective independence (EI) method is one of the most widely used OSP techniques, which uses the diagonal sum of the Fisher information matrix (FIM) to optimize the linear independence of the mode shape [2]. The EI algorithm calculates the OSP by maximizing both spatial independence and signal strength. The EI method proposed by Kammer [2,3] is an iterative method wherein the contributions of all predicted sensor positions are evaluated based on linear independence, and positions with small contributions are sequentially removed until they match the initial number of sensors. The final sensor layout maximizes the diagonal sum (trace) and determinant, and minimizes the condition number of the FIM. Maximizing the FIM results in a minimum covariance matrix estimation error. Jiang et al. [4] presented a modified EI method to insert the concept of FIM maximization and the re-orthogonalization of modal shapes through QR decomposition. Yang and Lu [5] reported that the final sensor layout can be established by eliminating candidate sensors in each iteration using the interval FIM obtained via the interval analysis technology.

The modal kinetic energy (MKE) method provides information on the DOFs for each mode to exhibit the maximum responses. Zhou et al. [6] proposed a triaxial accelerometer-based MKE3 OSP algorithm based on the relationship between the EI and MKE methods. Heo et al. [7] proposed a kinetic energy optimization technique to design the number of sensors and their locations and compared it with the EI method. Li et al. [8] investigated the relationship between the MKE and EI methods. Papadopoulos and Garcia [9] proposed two OSP methods based on Gram–Schmidt orthogonalization and principal component analysis with model reduction. Cruz et al. [10] provided a genetic algorithm for the OSP by evaluating the modal parameters and simulating different scenarios. Extending the EI concept, Hemez and Farhat [11] proposed an OSP algorithm based on the strain energy distribution. 

The modal assurance criterion (MAC) is a method that determines the OSP from a value by matching the mode shape obtained using the finite element model and the mode shape obtained from the experiment. The MAC-based OSP algorithm captures the maximum value from the elements on the off-diagonal line of the MAC matrix. He et al. [12] presented a modified modal assurance criterion (MMAC) to improve the modal energy at selected locations and an adaptive genetic algorithm for enhancing the computational efficiency. Fu and Yu [13] obtained the sensor numbers using MAC and the sensor location via a single parenthood genetic algorithm. Brehm et al. [14] enhanced the conventional MAC algorithm using an energy-based criterion. Coote et al. [15] compared several OSP techniques for the optimal measurement of a helicopter fuselage. 

Singular value decomposition (SVD) provides singular values and corresponding orthonormal singular vectors. Kammer [16] presented an OSP algorithm by maximizing the magnitudes of the singular values of the block Hankel matrix. Cherng [17] analyzed existing OSP methods and improved their performance for the modal parameters based on the SVD for a candidate-blocked Hankel matrix.

Existing OSP methods based on mode shape and frequency are inefficient because they only use low-frequency regional data. A frequency-effective independence technique based on the frequency domain using principal components evaluated from the frequency response function to obtain data in the low- and middle-frequency domains was presented by Rao et al. [18]. 

In this study, we propose an optimal sensor design method for an order-reduced system using constraint conditions, constrained dynamic characteristics, and the modal analysis theory. The constraint conditions are extracted by dividing the target modes into master and slave modes and establishing the relationship between them. By utilizing the reduced model of master DOFs, existing OSP methods, such as the modal reduction (MR) method based on the EI, MAC, MKE, and modal strain energy (MSE) methods, are modified. A constrained dynamic equation (CDE) approach using SVD and EI methods is presented, which is derived based on the corresponding OSP criteria. The iteration continues until it matches the initial number of sensors. The final sensor layout is composed of master DOFs with high signal strength. The validity and comparison of the proposed methods are illustrated in a numerical example of the OSP of a truss structure, which shows that the sensors are positioned at different locations because they are sensitive to the OSP algorithm criteria for meeting the objectives of measurement.

## 2. Formulations

The OSP algorithms were derived based on the relationship of the modal data. The incomplete target modal data were divided into slave and master modes, and the constraint condition between these two sets was established. Existing OSP algorithms, such as EI, FIM, MKE, MSE, SVD, and MAC techniques, were modified using the order-reduced system under the constraint condition. The modified OSP methods trace potent sensor locations by repeatedly deleting the DOFs corresponding to the lowest signal strength in the master DOFs at each iteration process. 

The OSP process begins with a modal analysis of the dynamic system. The dynamic equation to describe the free vibration of an N-DOF undamped dynamic system can be written as
(1)$$M\ddot{u}+Ku=0$$
where **M** and **K** denote the N×N mass and stiffness matrices, respectively, and **u** and $\ddot{u}$ denote the N×1 generalized displacement and acceleration vectors, respectively. The mathematical model is formulated using the modal characteristics of the natural frequency $\omega_{i}\left(i=1,2,\ldots ,N\right),$ and the corresponding normalized mode shape vector $\Phi_{i}\left(i=1,2,\ldots ,N\right)$. $\Phi_{i}$ denotes the *i*-th column of the mode shape matrix. 

It is difficult to collect a full set of mode-shaped data of the system. Incomplete target modes smaller than the system DOFs were considered to describe the dynamic responses. Assuming the first r(r<N) target modes, the transformation of the N×1 generalized coordinate vector **u** in terms of r×1  modal coordinate vector y and N×r mode shape matrix Φ is written as
(2)$$\left[\begin{array}{c} u_{s}\\ u_{m} \end{array}\right]\;=\left[\begin{array}{c} \Phi_{s}\\ \Phi_{m} \end{array}\right]y$$
where **y** is the r×1 modal displacement vector and r indicates the number of target modes. The subscripts *s* and *m* denote the slave and master modes, respectively. $\Phi_{s}$ and $\Phi_{m}$ denote the s ×r slave mode shape matrix and (N−s)×r master mode shape matrix, respectively, and (N−s) represents the number of candidate sensor locations. Here, (N−s)≥r.

The method gradually traces the OSP layout at the master DOF iteratively. The reduction process of the master modes continues until the number of candidate sensors corresponds to the initial sensor number, $N_{1}$ $\left(r\leq N_{1}\leq N\right)$. In the following approaches, the existing mode-shape-based OSP methods are modified by extending the concept of the constraint condition of Equation (2)

### 2.1. Modal Reduction–Effective Independence (MR–EI) Approach

This method is an OSP algorithm that synthesizes an order-reduced system using the EI method. When comparing the signal strength of the order-reduced system expressed by the master DOFs, the candidate DOF with low contribution is relocated to the slave DOF at each iteration. Thus, the candidate sensor locations gradually decrease and coincide with the number of locations in the master mode. Using the generalized inverse in the second equation of Equation (2) and solving it with respect to y yields
(3)$$y=\Phi_{m}^{+}\;u_{m}$$
where ‘+’ denotes the generalized inverse. The substitution of Equation (3) into the first equation of Equation (2) leads to
(4a)$$u=\Phi \Phi_{m}^{+}u_{m}$$
(4b)$$\mathrm{Or}\left[\begin{array}{c} u_{s}\\ u_{m} \end{array}\right]=\left[\begin{array}{c} \Phi_{s}\Phi_{m}^{+}u_{m}\\ u_{m} \end{array}\right]$$

The displacements in the slave DOFs are expressed in the master DOFs using the first equation of Equation (4b). The estimated displacement responses containing the external noise can be written as
(5)$$\hat{u}=\Phi \Phi_{m}^{+}\left(u_{m}-\eta_{m}\right)$$
where $\hat{u}$ denotes the predicted displacement vector and $\eta_{m}$ is the Gaussian white noise vector corresponding to the master DOFs. The noise effect in the slave DOFs is neglected, which affects the selection process of the OSPs. By pre-multiplying both sides of Equation (5) by **R**, the displacements at the master DOFs can be determined, where **R** denotes the Boolean matrix to define the master DOFs.

The FIM was utilized to obtain the covariance matrix for predicting the maximum expectations as the OSP criterion of the MR approach. The error covariance between the unbiased and biased displacement vectors, **u**, of Equation (4a)**,** and $\hat{u}$ of Equation (5), respectively, is expressed as
(6)$$J=\left[\left(u-\hat{u}\right){\left(u-\hat{u}\right)}^{T}\right]\;=\left(\Phi \Phi_{m}^{+}\eta_{m}\right){\left(\Phi \Phi_{m}^{+}\eta_{m}\right)}^{T}={\left[\frac{1}{\eta_{m}\eta_{m}^{T}}{\left(\Phi \Phi_{m}^{+}\right)}^{T}\left(\Phi \Phi_{m}^{+}\right)\right]}^{+}\;={\left(\frac{1}{\eta_{m}\eta_{m}^{T}}\right)}^{+}F^{+}$$
where F represents the FIM, $F={\left(\Phi \Phi_{m}^{+}\right)}^{T}\left(\Phi \Phi_{m}^{+}\right)$. The maximum FIM, as an objective function, minimizes the covariance matrix. This indicates that the estimated displacement vector $\hat{u}$ nearly coincides with the unbiased displacement vector u at the maximum FIM. If the inconsistency between the actual and estimated displacements is high, the corresponding DOFs yield a low contribution to the OSP; thus, these DOFs must be removed from the master DOFs. The removed row in the master DOFs is relocated to the slave mode shape matrix. The DOFs of slave modes do not have a significant impact on the OSP. The same process was repeated until the final sensors reached the initial number of sensors.

The EI method investigates the contribution of candidate sensor locations depending on the spatial independency. Thus, the contribution of each row of the master modes to the total eigenvalues of the FIM is compared, and the row with a low influence index is moved to the slave DOF. In 1990, Kammer presented an effective information distribution (EID) matrix written as
(7)$$E_{d}=\left[\xi \chi \right]⨂\left[\xi \chi \right]\lambda^{-1}$$
where $\xi =\left(\Phi \Phi_{m}^{+}\right)$, and χ and λ are the eigenvector and eigenvalue of the FIM ${\left(\Phi \Phi_{m}^{+}\right)}^{T}\left(\Phi \Phi_{m}^{+}\right)$, respectively. ⨂ denotes the term-by-term matrix multiplication. 

Using the EI method, the sensor placements are judged by the evaluation of the signal strength, wherein the eigenvalues affect the candidate DOFs. The importance of sensor placement is compared with the elements in the EI matrix. The sum of squares of each row of the $E_{d}\;$ matrix as an objective function represents the degree of contribution to the sensor placement, and is written as
(8)$$p_{j}=E_{d,j}E_{d,j}^{T},j=1,2,\ldots \left(N-s\right)$$
where the subscript *j* denotes the row of the *j*-th matrix $E_{d}$. 

The DOF in the candidate sensor locations corresponding to the lowest p is moved to the slave DOF at each iteration. The iteration was repeated until the number of candidate sensors coincided with the prescribed number of sensors.

### 2.2. CDE–EI Approach

This approach was derived by applying the SVD algorithm to a CDE using modal constraints. The dynamic responses of the order-reduced system can be predicted using incomplete modes as constraint conditions. The dynamic responses were described in accordance with the constraint conditions. From the first equation in Equation (4b), the following relationship between the master and slave displacement DOFs is established:(9)$$\left[\begin{array}{cc} I & -\Phi_{s}\Phi_{m}^{+} \end{array}\right]\left[\begin{array}{c} u_{s}\\ u_{m} \end{array}\right]=0$$
where **I** denotes the s×s identity matrix.

Equation (9) represents the constraints that restrict the dynamic responses of the system. Thus, the system can be described by (N−s) dynamic equations. A more accurate response can be obtained with an increase in the number of slave DOFs because the number of constraints increases, from which the displacements can be more accurately described. As a result, the responses reflecting the modal characteristics at a full set of DOFs can be predicted.

Utilizing Equations (1) and (9) in the dynamic equation of motion for a constrained dynamic system derived by Udwadia and Kalaba [19] yields
(10)$$\ddot{u}=\ddot{\hat{u}}+M^{-1/2}{\left(AM^{-1/2}\right)}^{+}\left(b-A\ddot{\hat{u}}\right)$$
where $\ddot{\hat{u}}=-M^{-1}Ku$ from Equation (1), matrix A is the coefficient matrix on the left-hand side of Equation (9), A**=**$\left[\begin{array}{cc} I & -\Phi_{s}\Phi_{m}^{+} \end{array}\right]$, and b is the right-hand-side term in Equation (9), b=0. 

Equation (10) is modified by the difference between the constrained and unconstrained acceleration vectors as follows:(11a)$$\ddot{u}-\ddot{\hat{u}}=M^{-1/2}{\left(AM^{-1/2}\right)}^{+}A\left(\delta \ddot{u}\right)$$
(11b)$$\mathrm{or}\left[I-M^{-1/2}{\left(AM^{-1/2}\right)}^{+}A\right]\delta \ddot{u}=0$$
where $\ddot{u}=\ddot{u}-\ddot{\hat{u}}$.

As $\delta \ddot{u}\neq 0$, the dynamic characteristics of the system were obtained using SVD, which can be applied to the rank-deficient matrix $\left[I-M^{-1/2}{\left(AM^{-1/2}\right)}^{+}A\right]$ to evaluate its contribution to the sensor placement. The rank-deficient matrix Q is decomposed as follows:(12)$$Q=U\Sigma V^{T}$$
where U and V are real orthogonal matrices, and Σ is a diagonal matrix with a non-negative real. 

SVD is related to the eigenvalue decomposition. The non-zero elements of matrix Σ correspond to the square root of the non-zero eigenvalue of $Q^{T}Q$ or $QQ^{T}$, and its rank corresponds to the rank of matrix Q. Vectors **V** and **U** are eigenvectors of $Q^{T}Q$ or $QQ^{T}$. By inserting the eigenvectors χ of $Q^{T}Q$ or $QQ^{T}$ and $\xi =Q=I-M^{-1/2}{\left(AM^{-1/2}\right)}^{+}A$ into Equation (7), the EI method based on the CDE approach can be iteratively applied, and the objective function of Equation (8) can be obtained. The DOF with low contribution to the EID is relocated to the slave DOFs. The final sensor locations are obtained by iterating the same process as the EI method until they match the initial number of sensors.

### 2.3. Modified MKE Method

The modified MKE (MMKE) method evaluates the degree of dynamic contribution in an entire finite element model. Using Equation (2), the MMKE matrix is modified in terms of the target modes and the mass matrix as follows.
(13)$$P={\left(\Phi \Phi_{m}^{+}\right)}^{T}M\left(\Phi \Phi_{m}^{+}\right)$$

Matrix P is expressed by the (N−s)×(N−s) matrix of the master DOFs. The sum of squares of each row of the MMKE matrix is utilized as an objective function to evaluate the contribution to the sensor placement.
(14)$${\mathrm{MMKE}}_{j}=P_{j}P_{j}^{T},j=1,2,\ldots ,N-s$$
where $P_{j}$ indicates the *j*-th row of the MMKE matrix. ${\mathrm{MMKE}}_{j}$ is evaluated to measure the signal strength, and the DOF with the smallest value of ${\mathrm{MMKE}}_{j}$ is relocated to the slave mode at each iteration. By repeating the same process, the final master DOFs can be distinguished from the sensor layout.

### 2.4. Modified MSE Method

The modified MSE (MMSE) method utilizes the stiffness matrix instead of the mass matrix in Equation (13) using the MMKE method. The following objective function is evaluated by the sum of squares of the rows of the MMSE matrix: (15)$$P={\left(\Phi \Phi_{m}^{+}\right)}^{T}K\left(\Phi \Phi_{m}^{+}\right)$$

The DOF corresponding to the smallest **P** is moved to the slave DOFs. The remainder of the procedure is similar to that of the MMKE method by iteration.

### 2.5. Modified MAC Method

Modified MAC (MMAC) measures the degree of linearity between two vectors of mode shapes and depends on the angle expressed by the dot product between the two vectors. By substituting Equation (2) into the dynamic equation of Equation (1) and pre-multiplying the result by $\Phi^{T}$, we obtain
(16)$$\Phi^{T}M\Phi \ddot{y}+\Phi^{T}K\Phi y=0$$

Utilizing $y=\Phi_{m}^{+}u_{m}$ of Equation (3) in Equation (16) and pre-multiplying its result by ${\left(\Phi_{m}^{T}\right)}^{+}$ yields
(17)$${\left(\Phi \Phi_{m}^{+}\right)}^{T}M\left(\Phi \Phi_{m}^{+}\right){\ddot{u}}_{m}+{\left(\Phi \Phi_{m}^{+}\right)}^{T}K\left(\Phi \Phi_{m}^{+}\right)u_{m}=0$$

Equation (17) is expressed by (N−s) dynamic equations corresponding to the master DOFs in the time domain. By transforming the dynamic equation of Equation (17) to the characteristic equation by eigensolutions and solving the resulting equation, we obtain the natural frequency and the corresponding mode shape matrix.

The value in the MMAC matrix determinies the correlated mode pairs and ranges between 0 and 1. The MMAC matrix can be written as
(18)$$MMAC=\frac{{\left(\psi_{i}^{T}\psi_{j}\right)}^{2}}{\left(\psi_{i}^{T}\psi_{i}\right)\left(\psi_{j}^{T}\psi_{j}\right)}$$
where $\psi =\Phi \Phi_{m}^{+}$. 

If the MMAC value is 1, the two vectors match, and if the value is zero, they are orthogonal. The diagonal elements in the MMAC matrix take a value of 1 because the two mode vectors match and the off-diagonal element values are less than 1. A small MMAC value indicates a clear distinction between the two modes because the relationship between the two modes is small. A large value indicates that the distinction is unclear. The MMAC-based OSP technique optimizes the sensor layout corresponding to the minimum off-diagonal elements to be easily distinguished.

The row corresponding to the maximum MMAC value of the off-diagonal in the MMAC matrix is deleted and relocated to the slave DOFs. The objective function to evaluate the MMAC value between the two mode shape vectors can be written as
(19)$$p=\sum_{\begin{array}{c} i=1,j=1\\ i\neq j \end{array}}^{N-s}MAC_{ij}^{2}.$$

The MMAC values in Equation (18) and their evaluation using Equation (19) are iteratively calculated, and the row with a high *p* value is relocated to the slave DOFs until the number of master DOFs coincides with the optimal sensor number.

## 3. Numerical Example

The validity of the proposed method was illustrated in the truss structure presented by Sun and Buyukozturk [20]. The truss structure in Figure 1 comprised 27 members and 15 nodes, and was simply supported. Each node had two DOFs owing to the horizontal and vertical displacements. The truss structure had 27 displacement DOFs, except for the boundary conditions. The material properties of the truss members are as follows: elastic modulus E=200GPa, mass density $\rho =7860\mathrm{kg}/m^{3}$. The elements 1, 2, 4, 6, 8, 10, 12, 14, 16, 18, 20, 22, 24, 26, and 27 have a cross-sectional area of $0.01\;m^{2}$, and the elements 3, 5, 7, 9, 11, 13, 15, 17, 19, 21, 23, and 25 have a cross-sectional area of $0.005\;m^{2}$.

This example specifies eight sensor locations out of 27 DOFs using the proposed methods and utilizes the lowest six mode shape matrices as the target modes. The initial candidate sensor placements can be selected based on experience and structural topology. In this numerical example, an initial sensor in the slave DOFs was taken as the horizontal-displacement DOF at node 15 by the MMKE method. The sensor location with the lowest MMKE contribution was classified as the slave DOF, and the remainder as master or candidate DOFs. 

The final sensor layouts predicted by the proposed OSP algorithm are summarized in Table 1 and Figure 2. In the first numerical experiment, the final sensor layout depending on the selection of the first slave DOF was compared using the MMKE method. The horizontal displacements at 15x and 7x were the first slave DOF. Here, the numbers denote the node, and x and y are the horizontal and vertical DOFs, respectively. Table 1 shows that the selection of the initial sensor location rarely affected the final sensor layout, although there was a difference of 13x and 15x of the OSPs. Thus, the other OSP algorithms in subsequent numerical experiments were performed with an initial DOF of 15x as the slave DOF.

The sensor layouts did not coincide with the results of Sun and Buyukozturk. The proposed methods also did not provide similar results, except for a few locations, as shown in Table 1. The OSP designed a suitable representative sensor set depending on the measurement purpose of the truss structure. The OSP technique for subsequent analysis must be chosen based on the algorithm criteria. The sensor locations were predicted to be distributed globally, rather than locally, over the structure. The truss structure in this example had left–right symmetry. If it was structurally symmetrical and the fundamental mode was dominant, the sensors may have been clustered on one side of the symmetry because it was possible to estimate the responses on the other side. 

The proposed OSP algorithms provided different locations depending on the sensor design criteria, as shown in Table 1 and Figure 2. It was observed that the sensors were positioned at the DOFs to represent the independent signal strength in accordance with the prescribed OSP criterion. However, considering the symmetrical structure, it was estimated that the sensor positions obtained by the proposed algorithms were uniformly distributed along the truss. The other algorithms, except for MR and MMSE, provided a sensor layout with four sensors evenly distributed in the horizontal and vertical DOFs owing to their equivalent contribution degree. It was observed that the MR method produced a sensor design biased in the vertical DOFs because the effect of noise on the slave DOFs was neglected when deriving Equation (6). In addition, the MMSE algorithm rarely provided biased sensor locations in the vertical displacement DOFs. Biased results were obtained because the MMSE algorithm governed the static responses using only the fundamental mode.

The measurements were performed for a variety of purposes, such as system identification, damage detection, vibration control, and data expansion. Thus, it can be concluded that the OSP design criteria meet the prescribed purpose for subsequent analysis using sensor measurement data.

## 4. Conclusions

In this study, we proposed OSP algorithms to design a sensor layout by deleting candidate DOFs that deviate from the objective function. The algorithms were derived using an order-reduced model with the constraint conditions of modal coordinates and modal analysis theory. The sensors established by different criteria were positioned at the DOFs to represent the independent signal strength. It was observed from the numerical example of the OSP of the truss structure that the proposed methods provided different locations depending on the contribution degree of the objective function utilized for system identification, damage detection, vibration control, and data expansion. The inconsistency in the OSP design resulted from the different OSP algorithm criteria. Thus, the sensors must be designed such that they meet the prescribed measurement purpose for subsequent analysis using measurement data.

## Figures and Tables

**Figure 1 sensors-22-00589-f001:**
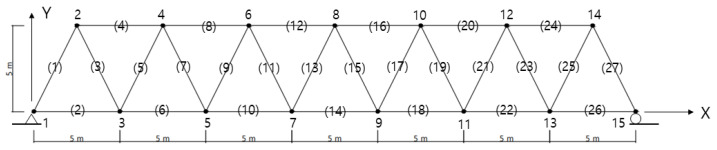
A plane truss. The number without parentheses indicates the node and the number with parentheses the element.

**Figure 2 sensors-22-00589-f002:**
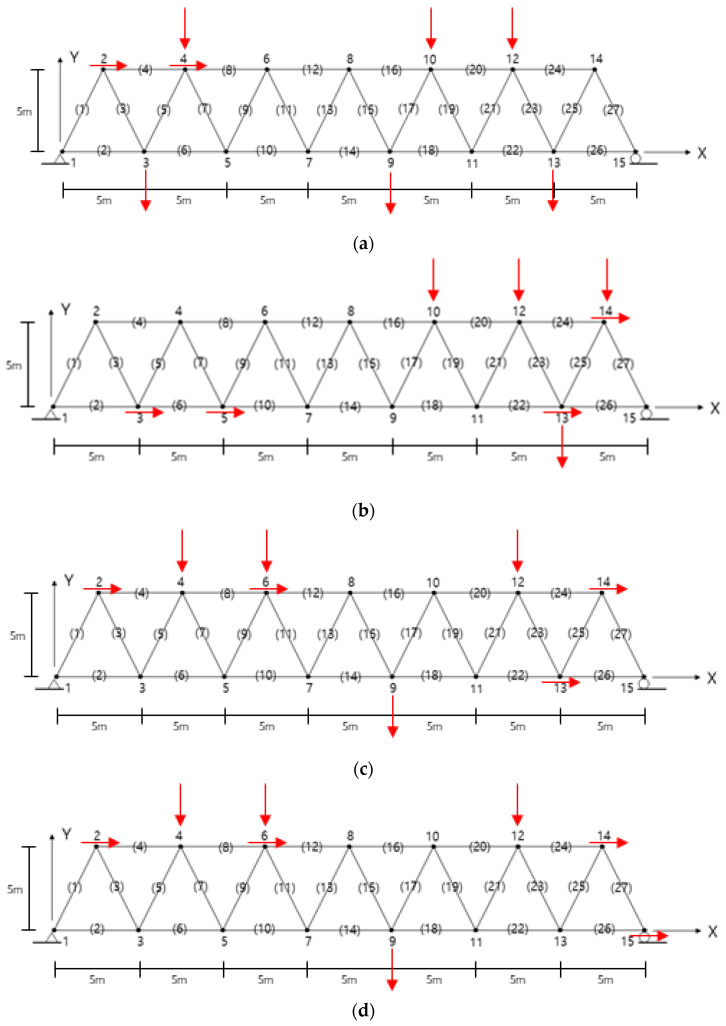
OSPs obtained from the proposed methods: (**a**) MR–EI, (**b**) CDE–EI, (**c**) MMKE27, (**d**) MMKE11, (**e**) MMSE, (**f**) MMAC, (**g**) Hao. The number without parentheses indicates the node and the number with parentheses the element.

**Table 1 sensors-22-00589-t001:** Sensor layouts according to the OSP algorithms proposed in this work.

	Horizontal Node	Vertical Node	OSP Criterion	Objective Function
MR-EI	2, 4	3, 4, 9, 10, 12, 13	Difference between unbiased and biased displacement data	Equation (8)
CDE-EI	3, 5, 13, 14	10, 12, 13, 14		
* MMKE 27	2, 6, 13, 14	4, 6, 9, 12	Modal kinetic energy	Equation (14)
* MMKE 11	2, 6, 14, 15	4, 6, 9, 12		
MMSE	2, 4, 13	3, 6, 7, 9, 13	Modal strain energy	Equation (14)
MMAC	4, 5, 8, 14	2, 5, 6, 7	Matching degree of mode shape vector	Equation (19)
Sun and Buyukozturk	2, 7, 12, 14	4, 7, 10, 13		

* The numbers of MMKE indicate the initial slave DOF.

## Data Availability

The data used to support the findings of this study are included within the article.

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
