# Peer review of "Optimal Sensor Placement in Reduced-Order Models Using Modal Constraint Conditions"

_sensors, 2022, doi:10.3390/s22020589_

Round 1

Reviewer 1 Report

The paper presents an approach for optimal sensor placement that can be of interest in SHM. The method is correclty presented and illustrated in a simple numerical example. In order for the paper to be acceptable the authors are advised to consider the following points:

  • please clearly state the novelty of your method with respect to existing methods in the literature, as well as its main limitations (e.g. you need a model of the structure)
  • please compare the optimal sensor locations you get with your method with the results of other well established techniques existing in the literature
  • please discuss in the intrdoduction in more details why an OSP method is useful in SHM especially to better frame your paper within the special issue for which is being candidated
  • MMAC seems the same as MAC. Please specify

Author Response

The paper presents an approach for optimal sensor placement that can be of interest in SHM. The method is correctly presented and illustrated in a simple numerical example. In order for the paper to be acceptable the authors are advised to consider the following points:

A) Thank you for your invaluable comments. I am well aware of the adequacy or shortcomings of the results of this study. What is lacking is that we plan to conduct more in-depth research by using it as a future research subject. 

please clearly state the novelty of your method with respect to existing methods in the literature, as well as its main limitations (e.g. you need a model of the structure)

A) Measuring methods by sensors are used for a variety of purposes such as structural health monitoring, damage detection, and control, etc. This study proposed the optimal sensor layout design method according to the purpose of use, either theoretically or through numerical experiments. The proposed methods are highlighted by the model reduction approach using constraint condition of modal equation unlike the existing methods. An integrated design technique that satisfies all objectives should be devised through analysis of the optimal sensor layout design results depending on the OSP approaches. These contents are explained in the manuscript.

  • please compare the optimal sensor locations you get with your method with the results of other well established techniques existing in the literature

A) The analysis results of the methods proposed in this study and the method proposed by Sun and Buyukozturk were compared through a numerical example (Table 1). The difference between these test results is due to the measurement purpose as indicated in the previous comment.

  • please discuss in the introduction in more details why an OSP method is useful in SHM especially to better frame your paper within the special issue for which is being candidate.

A) We mentioned the need for this study as follows.

“It is impractical to measure the response at a full set of DOFs to electromagnetic induction in the finite element analysis model. Measurement should be made by sensors that are smaller than system order. Optimal sensor design is to establish a limited number of sensors and sensor positions in order to investigate the performance of the structure.” This work was performed based on this regard. The ultimate purpose of this study was established as follows.

“This study proposes optimal sensor design method of order-reduced system using constraint conditions, constrained dynamic characteristics, and modal analysis theory.”

  • MMAC seems the same as MAC. Please specify

A) This study was carried out by modifying the existing methods and introducing the constraint condition of modal equation, so that M in front of all proposed methods was added.

Reviewer 2 Report

The authors discuss one of the important issues, the number of sensors for real applications. The authors proposed an optimal sensor placement algorithm considering the constraint condition between the master and slave modes. However, the presented study with a truss structure cannot demonstrate the usability of the proposed algorithm. The reviewer suggests the authors provide a simulation and compare with the results based on the proposed algorithm. The number of sensors is actually based on the algorithm criterion. The authors should discuss more as this can be different for different structures. Additionally, the quality of the current form both text and figures is very low, and there are some mistakes and typos.

Author Response

The authors discuss one of the important issues, the number of sensors for real applications. The authors proposed an optimal sensor placement algorithm considering the constraint condition between the master and slave modes.

A) Thank you for your invaluable comments. I am well aware of the adequacy or shortcomings of the results of this study. What is lacking is that we plan to conduct more in-depth research by using it as a future research subject.

1) The presented study with a truss structure cannot demonstrate the usability of the proposed algorithm.

A) The beam member was considered as an example of analysis, but the truss was selected as a structure rather than a beam member. A truss structure is a structure composed of axial members and nodal joints. It was chosen as an example structure because it is not a simple structure that the load transfer path at the node changes discontinuously. Measuring methods by sensors are used for a variety of purposes such as structural health monitoring, damage detection, and control, etc. This study proposed the optimal sensor layout design method according to the purpose of use, either theoretically or through numerical experiments. The proposed methods are highlighted by the model reduction approach using constraint condition of modal equation unlike the existing methods. An integrated design technique that satisfies all objectives should be devised through analysis of the optimal sensor layout design results depending on the OSP approaches. These contents are explained in the manuscript.

2) The reviewer suggests the authors provide a simulation and compare with the results based on the proposed algorithm.

A) The analysis results of the methods proposed in this study and the method proposed by Sun and Buyukozturk were compared through a numerical example (Table 1). The difference between these test results is due to the measurement purpose.

3) The number of sensors is actually based on the algorithm criterion.

A) The number of sensors should be initially assumed.

4) The authors should discuss more as this can be different for different structures.

A) We agree that application examples are necessary for various structures to verify the validity of the results of this study although we covered the example for the truss. Such verification experiments will be conducted in future studies.

5) Additionally, the quality of the current form both text and figures is very low, and there are some mistakes and typos.

A) As you pointed out, the low quality of the drawings is acceptable. I think that the quality was low as a result of reducing the size of the figure in a limited space. Since the figures are saved in the CAD file, we can promise to submit the figures with greatly improved quality in the final version. In addition, the English of the text has been revised by requesting the Elsevier Editing Service, and additional review will be conducted.

Round 2

Reviewer 1 Report

The paper can be accepted for publication after the revision.